# Assessing the Impact of Non-Pharmaceutical Interventions on Consumer Mobility Patterns and COVID-19 Transmission in the US

**DOI:** 10.3390/ijerph21010067

**Published:** 2024-01-07

**Authors:** Joseph Zuccarelli, Laura Seaman, Kevin Rader

**Affiliations:** 1The Charles Stark Draper Laboratory, Cambridge, MA 02139, USA; lseaman@draper.com; 2Department of Statistics, Harvard University, Cambridge, MA 02139, USA; rader@stat.harvard.edu

**Keywords:** COVID-19, social distancing measures, mobility, mixed-effects modeling, longitudinal study, policy analysis

## Abstract

The initial outbreak of COVID-19 during late December 2019 and the subsequent global pandemic markedly changed consumer mobility patterns worldwide, largely in response to government-ordered non-pharmaceutical interventions (NPIs). In this study, we investigate these changes as they relate to the initial spread of COVID-19 within two states—Massachusetts and Michigan. Specifically, we use linear and generalized linear mixed-effects models to quantify the relationship between four NPIs and individuals’ point-of-sale (POS) credit card transactions, as well as the relationship between subsequent changes in POS transactions and county-level COVID-19 case growth rates. Our analysis reveals a significant negative association between NPIs and daily POS transactions, particularly a dose–response relationship, in which stringent workplace closures, stay-at-home requirements, and gathering restrictions were all associated with decreased POS transactions. We also uncover a significant positive association between 12-day lagged changes in POS transactions compared to pre-pandemic baselines and county-level COVID-19 case growth rates. Overall, our study supports previous findings that early NPIs reduced human mobility and COVID-19 transmission in the US, providing policymakers with quantitative evidence concerning the effectiveness of NPIs.

## 1. Introduction

In early January 2020, Chinese scientists confirmed that a new pneumonia-like illness identified at a food market in Wuhan, China was transmittable from human to human. Shortly after, three cases of the disease, now known as COVID-19 (SARS-CoV-2), were reported in Japan and Thailand, causing the Centers for Disease Control and Prevention (CDC) to begin screenings at popular international airports within the US. On 20 January 2020, the CDC identified a resident of Washington state as the first person in the United States with a confirmed case of COVID-19 [1]. By the end of the month, the World Health Organization (WHO) was left with no choice but to declare a public health emergency, as the number of confirmed COVID-19 cases worldwide had grown above 9800, including over 200 confirmed deaths. Finally, with the number of confirmed cases and deaths continuing to rise over the entire month of February, the WHO officially declared COVID-19 a pandemic on 11 March 2020. Just two days later, former US President Donald Trump followed suit, declaring COVID-19 a national emergency [2].

COVID-19 is known to cause severe respiratory system damage as well as other potentially fatal symptoms. Although the death rate of COVID-19 is lower than that of other notable epidemics such as Ebola [3], COVID-19 is highly contagious, allowing it to kill more people than other deadlier diseases [4]. The contagious nature of COVID-19, in conjunction with extensive human mobility, enables the virus to have an extremely high rate of transmission. Therefore, one common approach used to combat the spread of COVID-19 is social distancing recommendations.

Social distancing is defined as a public health practice that aims to prevent infected individuals from coming into contact with healthy individuals to reduce opportunities for disease transmission [5]. This practice hinges on the basic concept that infected particles in the air are less likely to be transmitted with an increased distance between people [6]. Adherence to social distancing recommendations is pertinent to reducing the spread of COVID-19, as the disease is not only highly contagious but also infects many individuals without showing any common symptoms [7]. Unfortunately, however, voluntary social distancing guidelines are not sufficient to end the pandemic, as people may choose not to adhere to these recommendations, thus requiring that governments take concrete action to limit close contact between many people [8].

The rapid spread of COVID-19 throughout the US required a swift government response. The US federal government chose to defer responsibility for COVID-19 policy responses primarily to the states [9]. Prior to the development of vaccines, state governments relied heavily on various non-pharmaceutical interventions (NPIs) to control the disease spread. Popular types of NPIs included containment measures such as domestic or international travel bans, individual protection measures such as mask-wearing requirements, social distancing measures such as business closures and gathering bans, and health system measures such as testing and contact tracing [10]. Governors across all fifty states chose to implement these different forms of NPIs at various points over time, while learning from each other in the process [11].

In this study, we assess the impact of multiple NPIs on consumer mobility patterns and the initial outbreak of COVID-19 during 1 January through 31 May 2020 within urban regions in two states: Boston, MA, and Ann Arbor, MI. Unlike previous studies, our mobility measure is individuals’ daily point-of-sale (POS) credit card transaction counts. POS transactions likely represent moments of high virus exposure, as store checkout lines are high-traffic areas where individuals often come into close contact with others, making them a valuable mobility measure. Using linear and generalized linear mixed-effects regression models, we empirically investigate the relationship between NPIs and individual-level POS transactions along with subsequent changes in POS transaction patterns compared to pre-pandemic baselines and county-level confirmed COVID-19 case growth rates. Our results provide policymakers with quantitative evidence concerning the effectiveness of NPIs in terms of reducing consumer mobility and controlling the spread of COVID-19, which is pertinent given that these policies have far-reaching consequences. Although government intervention is essential in inhibiting the spread of COVID-19, NPIs widely disrupted daily life, leading to other unintended economic and public health consequences [12,13].

## 2. Related Work

Governments across the globe used a wide range of measures to control the COVID-19 outbreak. During the initial stages of the pandemic, the Chinese government placed Wuhan under strict lockdown orders, and travel bans between China and other countries immediately took effect in an effort to prevent the nationwide and global spread of the virus. Kraemer et al. [14] used generalized linear models to investigate the relationship between human mobility and COVID-19 transmission in Chinese cities, finding that government control measures effectively mitigated the virus spread. Additionally, using a similar modeling approach, Zhao et al. [15] identified a positive association between the quantity of domestic passenger travel and confirmed COVID-19 cases within ten cities outside of Hubei province. Moreover, a comparable study by Tian et al. [16] also used regression analysis, finding that travel restrictions in Wuhan City were associated with the delayed arrival of COVID-19 in other Chinese cities by 2.91 days. Furthermore, Chinazzi et al. [17] used a global disease transmission model to explore the relationship between the Wuhan travel quarantine and the COVID-19 spread. They found that this restriction slowed COVID-19 progression by three to five days and contributed to a 77% decrease in COVID-19 cases exported from mainland China.

As many other countries started to experience COVID-19 outbreaks, NPIs became more widely adopted. Most notably, the Kingdom of Saudi Arabia (KSA) quickly controlled the spread of COVID-19 via the implementation of several NPIs, such as travel suspensions, local curfews, closures of public spaces, and limits on gatherings [18]. AlJohani and Mutai [19] used stochastic models to assess the impact of NPIs on COVID-19 spread, finding that the unique and aggressive implementation of NPIs in the KSA contributed substantially to the downward trend of the disease spread. Additionally, Perez-Saez et al. [20] employed a similar SEIR-type modeling approach to investigate the association between NPIs and COVID-19 transmission in the KSA during the initial disease outbreak. They uncovered strong associations between reductions in transmission and several NPIs, specifically school closures, curfews, and lockdown orders. Moreover, Bisanzio et al. [21] evaluated the effect of NPIs on the spread of COVID-19 using a spatio-temporal individual-based model. They found that mask wearing, physical distancing, and contact tracing proved highly effective in controlling the disease outbreak within the KSA. Furthermore, Alhomaid et al. [22] used hybrid simulation modeling to measure the impact of NPIs implemented by the KSA government at the beginning of the pandemic. Their model suggests that COVID-19 cases would have increased 18-fold in the KSA without NPIs, highlighting the importance of early intervention to limit disease spread.

The US government also attempted to slow COVID-19 spread via the implementation of various NPIs. Zhang and Warner [23] examined the impact of several NPIs on changes in the daily COVID-19 infection growth rate across US states via an event-study design. They found that shutdowns and mask mandates significantly reduced the spread of COVID-19 immediately after being imposed statewide. Additionally, Siedner et al. [24] evaluated the effect of social distancing measures on initial state-level COVID-19 epidemic growth using an interrupted time-series analysis, finding that statewide policy interventions were associated with a decrease of 3090 cases after seven days, and 68,255 cases after 14 days. Moreover, Abouk and Heydari [25] applied difference-in-differences and event-study methodologies to evaluate the effect of NPIs on movement trends over time, discovering that statewide stay-at-home orders and limits on bars and restaurants were significantly associated with reduced human mobility. Furthermore, using a similar difference-in-differences study design, Dave et al. [26] found that the adoption of shelter-in-place orders was associated with a 9–10% increase (relative to the pre-treatment period) in the rate at which state residents remained in their homes full-time. They also discovered that approximately three weeks following intervention, cumulative COVID-19 cases fell by roughly 53%. Additionally, Li et al. [27] employed an event-study framework and found that policies which created incentives for less movement and prohibited large gatherings decreased mobility most significantly. More specifically, their work indicated that stay-at-home orders and workplace closures led to the greatest reductions in COVID-19 confirmed case growth rates during the initial outbreak.

A few other studies with similar aims carried out more granular analysis via the use of US county-level data. Using event-study regressions, Courtemanche et al. [28] found that social distancing measures reduced the confirmed COVID-19 case growth rate at the county level by 5.4% after one to five days, 6.8% after six to ten days, 8.2% after eleven to fifteen days, and 9.1% after sixteen to twenty days. More specifically, their work indicated that shelter-in-place orders and closure of public places were most effective in terms of slowing the virus spread. Moreover, Gupta et al. [29] employed a similar event-study design, finding that first case announcements, emergency declarations, and school closures reduced mobility by 1–5% after five days and 7–45% after twenty days (relative to the baseline period). Additionally, a similar study by Badr et al. [30] identified a strong correlation between mobility patterns and decreased COVID-19 case growth rates for the most affected US counties via generalized linear modeling. They also found that the impact of changes in mobility patterns, which dropped by 35–63% in comparison to normal conditions, on virus transmission were unlikely to be perceptible for 9–12 days. Furthermore, using a slightly different modeling approach, Jalali et al. [31] uncovered that the delayed implementation of public health interventions and a low level of compliance with stay-at-home orders, along with health disparities, significantly contributed to the initial spread of COVID-19.

Overall, previous research indicates that the early implementation of various NPIs effectively reduced human mobility and slowed COVID-19 spread. In this study, we expand upon prior work through the analysis of a novel, individual-level consumer mobility dataset that covers the initial COVID-19 outbreak. Using this dataset, we aim to characterize the effectiveness of NPIs in terms of controlling the disease spread within the US through reductions in consumer mobility. Previous studies achieved similar objectives, proposing intriguing relationships between government policy interventions, human mobility, and COVID-19. Some analyzed these relationships using state-level data, while others carried out more granular investigations at the county level. However, previous studies did not explore these concepts through the use of individual-level consumer data. Ultimately, this research aims to provide a thorough quantitative understanding of NPIs and their impacts, which can support future policy decisions concerning infectious diseases like COVID-19.

## 3. Data

### 3.1. Non-Pharmaceutical Intervention Data

The Oxford COVID-19 Government Response Tracker (OxCGRT) project systematically collected information on policy measures implemented by US government leaders to curb the spread of COVID-19. For each state, OxCGRT used publicly available sources such as news articles and government press releases to record data on indicators of government response to the outbreak of COVID-19 dating back to the beginning of 2020, with a specific focus on the period from 1 March to 31 May [32]. In our study, we focus on four of the NPIs (i.e., national emergency, workplace closures, stay-at-home requirements, and gathering restrictions) recorded by OxCGRT. Table 1 provides an in-depth description of each NPI variable. The full dataset is publicly available online via GitHub: https://github.com/OxCGRT/covid-policy-tracker (accessed on 15 June 2023).

### 3.2. Consumer Mobility Data

BDEX, a popular data exchange platform, privately made available POS transactions carried out by two random samples of 2000 credit lines from Boston, MA and Ann Arbor, MI during the time period from 1 January to 31 May 2020 [33]. POS transactions are likely moments of high virus exposure, as store checkout lines are high-traffic areas where individuals often come into close contact with others, making them a valuable mobility measure. Each observation within this dataset represents an individual account’s daily POS transaction count. In order to measure changes in consumer mobility, we compared POS transaction counts during the COVID-19 pandemic to normal shopping patterns observed during a pre-pandemic baseline period.

### 3.3. COVID-19 Data

County-level COVID-19 case data were obtained via each state government’s official website [34,35]. Both the MA and MI state governments maintained publicly available repositories listing daily and cumulative confirmed case counts during the time period from 1 March to 31 May 2020. Note that we only consider COVID-19 spread within two counties, Suffolk County, MA and Washtenaw County, MI, as these are the resident counties of the two cities for which we obtained consumer mobility data via BDEX. Given the aims of our study, we derived the daily exponential confirmed COVID-19 case growth rate (i.e., the natural logarithm of daily confirmed cases minus the natural logarithm of confirmed cases on the prior day, multiplied by 100 to be interpretable as a percent) for use in subsequent statistical analysis [36]. Note that we chose this functional form because epidemiological models predict exponential growth in the absence of intervention [37]. In line with similar studies, we added one to each daily case count to prevent the logarithm of cases from being undefined on days on which there were no cases in a given county [38].

### 3.4. Control Variables

Additional COVID-19-related control variables were considered in this study, specifically state-level counts of daily total tests administered (i.e., the sum of positive and negative tests on a given day). We included this parameter in our models to account for the fact that the number of confirmed COVID-19 cases was initially restricted by testing availability within each region. Although controlling for county-level testing would be more desirable, these data were not recorded during the initial disease outbreak. Despite this limitation of our dataset, controlling for state-level testing should help alleviate bias given that policy variation also occurred at the state level. Note that we represented testing as a rate (per 100,000 population) within our models to account for population differences between states.

## 4. Methods

There are two main features of the dataset under analysis in this study. First, data were collected for numerous individuals over time, resulting in a panel data structure. Second, multiple NPIs were instituted on different dates at different stringency levels (i.e., varying degrees of intensity) over various periods of time. Given these two features of our dataset, we utilized linear and generalized linear mixed-effects models with autoregressive covariance structures to investigate the following two research questions [39,40,41,42,43]:How did early government NPIs impact individual-level consumer mobility patterns?How did changes in individuals’ consumer mobility patterns impact the initial spread of COVID-19 within their residing county?

The general framework of our quantitative analysis consisted of the following two steps. First, we investigated the relationship between NPIs and individual consumer mobility patterns via generalized linear mixed-effects models. We provide detailed descriptions of these models within the Consumer Mobility Modeling subsection (see Section 4.1). Next, we explored the relationship between subsequent changes in each sample’s consumer mobility patterns and the spread of COVID-19 via linear mixed-effects models. We provide detailed descriptions of these models within the COVID-19 Modeling subsection (see Section 4.2).

### 4.1. Consumer Mobility Modeling

Given the nature of our consumer mobility data (i.e., daily POS transaction counts clustered at the individual level), we fit Poisson mixed-effects models to investigate the relationship between NPIs and POS transactions. Recall from the previous section that we focused on assessing the impact of four interventions (i.e., national emergency, workplace closures, stay-at-home requirements, and gathering restrictions) on the consumer mobility patterns of two samples of residents from Boston, MA and Ann Arbor, MI. Therefore, we fit a total of eight models—one for each NPI of interest, subset by consumer sample—formally defined as follows:(1)$$\log\left(transactions_{tj}\right)=\beta_{0}+\beta NPI_{t}+\beta weekend_{t}+\gamma_{0j}.$$

The response variable, $transactions_{tj}$, denotes the POS transaction count observed on date *t* for individual *j*. The indicator variable, $NPI_{t}$, represents one of the four interventions of interest in our study. Note that we treated each NPI as a categorical variable with multiple levels (see Table 1 for descriptions of each level). Therefore, fitting each separate model with four different NPI variables resulted in multiple β coefficients due to one hot encoding, a process used to convert categorical variables into multidimensional binary vectors to make the coefficient terms resulting from each model more easily interpretable. We controlled for seasonal patterns induced by the day of the week with the variable $weekend_{t}$. Similarly, we adjusted for individual-level effects with the term $\gamma_{0j}$, which represents the random intercept for individual *j*. We fit each model using a first-order autoregressive covariance structure with homogeneous variances to account for the time-series nature of our data.

### 4.2. COVID-19 Modeling

In order to investigate the relationship between changes in consumer mobility patterns and the spread of COVID-19, we took a slightly different approach than described in the previous section. Poisson models were no longer necessary as our response variable was not a daily count. Instead, we modeled the daily exponential growth rate in confirmed COVID-19 cases (i.e., the natural logarithm of daily confirmed COVID-19 cases minus the logarithm of daily confirmed COVID-19 cases on the prior day, multiplied by 100 to be interpretable as a percent) for the counties of our two resident samples (i.e., Suffolk County, MA and Washtenaw County, MI), making linear models more appropriate. Note that we chose this functional form because epidemiological models predict exponential growth in the absence of intervention [37]. In line with similar studies, we added one to each daily case count to prevent the logarithm of cases from being undefined on days on which there were no cases in a given county [38].

The explanatory variable of interest in our COVID-19 models was the percentage change in consumer mobility patterns. As indicated in the Related Work section (see Section 2), substantial efforts were made to create datasets that reflect human mobility change since the outbreak of COVID-19. In this study, we attempted to do the same by creating a variable to represent the percentage of change between each sample’s total daily POS transaction count and pre-pandemic baseline days. Note that pre-pandemic baseline days represent a normal value for each day of the week and are defined as the median value of the sample’s daily POS transaction count during the period from 1 January to 29 February 2020. Using baseline days enables us to account for variations in consumer spending behavior by day of the week in our models.

A major challenge in modeling the relationship between changes in mobility patterns and the outbreak of COVID-19 was determining the proper lag in time between mobility data and COVID-19 transmission. From the time an individual makes initial contact with the virus and contracts the disease, confirming the diagnosis is a complex process. There are several factors that can affect the length of time necessary for a COVID-19 case to be confirmed—incubation period, testing speed, and reporting delay, to name a few. Previous studies found that the COVID-19 incubation period can vary, yet most cases manifest themselves between 3 and 7 days after initial infection [44]. The lag caused by testing speed and delays in reporting is much more variable, as it is largely dependent upon the medical facilities within a region. Other statistical studies have attempted to quantify the optimal lag, most notably that by Badr et al. [30], who found that a lag of 9–12 days achieved the highest correlation between mobility and COVID-19 growth rate ratios for a single US all-county model.

In this study, we fit a total of four models—one for each mobility data lag ranging from 8 to 14 days–defined as follows:(2)$$case\;growth\;rate_{tj}=\beta_{0}+\beta_{1}\Delta transactions_{tjk}+\beta_{2}test\;rate_{tj}+\beta_{3}weekend_{t}+\gamma_{0j}.$$

The response variable, $case\;growth\;rate_{tj}$, denotes the daily confirmed COVID-19 case growth rate observed on date *t* for county *j*. The explanatory variable of interest, Δ$transactions_{tjk}$, represents the *k*-day lagged percentage of change in each sample’s daily total POS transaction count compared to baseline. We controlled for the state-level COVID-19 testing rate (per 100,000 population) and seasonal patterns induced by the day of the week (see variables $test\;rate_{tj}$ and $weekend_{t}$). Similarly, we adjusted for county-level effects with the term $\gamma_{0j}$, which represents the random intercept for county *j*. We fit each model using a first-order autoregressive covariance structure with homogeneous variances to account for the time-series nature of our data.

## 5. Results

### 5.1. Non-Pharmaceutical Interventions

Figure 1 displays the stringency of each NPI over time within our two states of interest—MA and MI. Table 2 displays the proportion of days spent by each state under each stringency level during our time period of interest (i.e., 1 January–31 May 2020). Refer back to Table 1 for in-depth descriptions of each NPI variable. On 13 March, President Trump declared COVID-19 a national emergency. Both state governments instituted NPIs around this time, yet the MI government was a bit faster to intervene and more active in terms of altering the intensity of each intervention.

In order to formally investigate whether the implementation of each NPI differed across states, we used chi-square tests. Workplace closures ($\chi^{2}=59.233$, p<0.01) were required in both states, yet the MA government required the closing of all but essential workplaces, while the MI government only required the closing of some sectors of work for a considerable duration. Stay-at-home requirements ($\chi^{2}=81.947$, p<0.01) were required in MI, yet only recommended in MA. Both state governments instituted strict gathering restrictions ($\chi^{2}=7.633$, p=0.05).

### 5.2. Consumer Mobility Modeling

Figure 2 displays the average individual POS transaction count over time across our two samples of residents (i.e., Boston, MA and Ann Arbor, MI). Qualitatively, the pattern of transactions by date shows similar consumer behavior. Immediately following the national emergency declaration on 13 March, the average individual POS transaction count rapidly declined, remaining exceptionally lower than normal until about late April into early May, when both consumer samples started to become noticeably more active again.

Using the *glmmTMB* package in R [45,46,47], we fit eight total Poisson mixed-effects models—one for each NPI of interest, subset by consumer sample—with first-order autoregressive covariance structures. Recall from Section 4.1 (Consumer Mobility Modeling) that these models are formally structured as follows:$$\log\left(transactions_{ij}\right)=\beta_{0}+\beta NPI_{i}+\beta weekend_{i}+\gamma_{0j}.$$
Table 3 displays the resulting fixed effect coefficients obtained from fitting each model on our two consumer samples (i.e., Boston, MA and Ann Arbor, MI, residents). Note that we were unable to acquire estimates for a few stringency levels of the various NPIs due to a lack of data.

The exponentiation of each model’s intercept term represents the average number of daily POS transactions over all individuals without any NPI on a weekday. Within each consumer sample, this estimate is very similar across the various NPIs, which follows intuitively considering that each NPI was instituted around the same date. However, across samples, it is evident that individuals from Ann Arbor, MI completed more daily POS transactions on average than individuals from Boston, MA.

The exponentiation of the NPI coefficient terms included in each model represents the multiplicative effect of instituting each NPI, controlling for the day of the week. Almost every exponentiated NPI term (with the exception of restrictions on gatherings between 101 and 1000 persons) is less than one and statistically significant at the α=0.05 level, indicating that each of these NPIs was associated with a decrease in the average number of daily POS transactions across individuals from both samples. Specifically, former US President Trump’s declaration of a national emergency was associated with a 10% reduction in the average daily POS transaction count among Boston residents, and a 24% reduction in the average daily POS transaction count among Ann Arbor residents. Required workplace closures were associated with a reduction of 17% in the average daily POS transaction count among Boston individuals, and a reduction of 42% in the average daily POS transaction count among Ann Arbor individuals. Recommended stay-at-home orders were associated with a 15% reduction in the average daily POS transaction count among Boston residents, and a reduction of 7% in the average daily POS transaction count among Ann Arbor residents. Required stay-at-home orders (not implemented in MA during the initial COVID-19 outbreak) were associated with a 30% reduction in the average daily POS transaction count among Ann Arbor residents. Restrictions on gatherings of less than 10 people were associated with a reduction of 15% in the average daily POS transaction count among Boston individuals, and a reduction of 30% in the average daily POS transaction count among Ann Arbor individuals.

Across all NPIs with multiple stringency levels (i.e., workplace closures, stay-at-home requirements, and gathering restrictions), as the stringency level increases, the NPI coefficient estimate decreases, indicating that both consumer samples engaged in fewer daily POS transactions on average as closures, requirements, and restrictions became more stringent. In epidemiological terms, this is considered a dose–response relationship. Hill’s criterion for causation states that “if a dose response is seen, it is more likely that the association is causal” [48,49], thus providing increased evidence of such a relationship between NPIs and consumer mobility, as measured by daily POS transaction counts.

### 5.3. COVID-19 Modeling

Figure 3 displays the daily confirmed COVID-19 case rate over time within the resident counties of our two consumer samples (i.e., Suffolk County, MA and Washtenaw County, MI). Across counties, both experienced pronounced spikes in the confirmed COVID-19 case rate. The spike within Washtenaw County occurred immediately following the declaration of COVID-19 as a national emergency. However, the spike within Suffolk County occurred closer to April and was much greater in terms of both duration and magnitude.

Figure 4 displays the percentage change in consumer mobility compared to pre-pandemic baselines (Δ transactions) over time across our two resident samples (i.e., Boston, MA and Ann Arbor, MI). Both samples exhibited massive reductions in POS transactions following the declaration of a national emergency on 13 March; however, the reduction among Ann Arbor residents was noticeably more intense. Neither consumer sample returned to normal in-person spending behavior until mid-May.

Using the *nlme* package in R [47,50], we fit four total mixed-effects models—one for each mobility data lag ranging from 8 to 14 days—with first-order autoregressive covariance structures. Recall from Section 4.2 (COVID-19 Modeling) that these models are formally structured as follows:$$case\;growth\;rate_{ij}=\beta_{0}+\beta_{1}\Delta transactions_{ijk}+\beta_{2}test\;rate_{ij}+\beta_{3}weekend_{i}+\gamma_{0j}.$$

Table 4 displays the resulting fixed effect coefficients obtained from fitting each model at mobility lags ranging from 8 to 14 days.

The intercept term included in each model represents the average daily confirmed COVID-19 case growth rate without any changes in consumer mobility and COVID-19 testing on a weekday. Across lag periods, the random MA county intercept is larger than the random MI county intercept, indicating that on average the daily confirmed COVID-19 case growth rate was larger in Suffolk County than Washtenaw County. The coefficient estimate of Δ transactions for each lag period represents the average change in the daily confirmed COVID-19 case growth rate associated with a 1% change in consumer mobility (as measured by POS transactions) compared to pre-pandemic baselines, controlling for the testing rate and the day of the week. Similar to Badr et al. [30], our results suggest that the strongest positive association between changes in consumer mobility and daily confirmed COVID-19 case growth rates appeared at a lag of 12 days. Specifically, a 10% percent decrease in the 12-day lagged change in consumer mobility was associated with a 3.2% decrease in the daily confirmed COVID-19 case growth rate.

## 6. Discussion

Our work is unique in comparison to other studies in that we conducted our analysis using a novel dataset documenting the consumer mobility patterns of two random samples of 2000 credit lines from Boston, MA and Ann Arbor, MI during the time period from 1 January to 31 May 2020. Most other analyses were carried out using aggregated nationwide mobility reports made publicly available by Google, SafeGraph, Teralytics, and Unacast. Using this exclusive dataset made privately available by BDEX, in conjunction with published NPI and COVID-19 datasets, we quantified the relationship between government-mandated NPIs and individual-level consumer mobility patterns (as represented by POS credit card transactions) along with the relationship between subsequent changes in consumer mobility patterns and county-level confirmed COVID-19 case growth rates.

Consumer mobility models detected the presence of a statistically significant negative association between individual-level POS transaction counts and all four NPIs of interest (i.e., national emergency, workplace closure, stay-at-home requirements, and gathering restrictions). Stringent workplace closures exhibited the strongest negative association with daily POS transactions. Specifically, required work closings were associated with a 17% reduction in the average daily POS transaction count among Boston residents and a 42% reduction among Ann Arbor residents. Across samples, the association between each NPI and POS transactions was more negative among Ann Arbor residents than Boston residents. Within samples, each NPI exhibited a dose–response relationship with POS transactions (i.e., incremental increases in the stringency of NPIs produced decreases in POS transactions), providing evidence of a causal relationship. These results support previous findings that the early implementation of NPIs, in particular workplace closures, stay-at-home requirements, and gathering restrictions, effectively reduced human mobility [25,26,27,29,31].

Our consumer mobility models possess a few limitations. First, several NPIs were implemented within a short period of time during the early stages of the pandemic, making it difficult to isolate the impact of each individual policy intervention on consumer mobility. Second, BDEX provided very little information about the consumers under observation in our dataset, thus creating some uncertainty concerning our study population. Third, although our models include several control variables and their results provide evidence of causality, we could not rule out all possible threats to causal inference, as is typical of observational studies. Possible confounders include informal encouragement from government officials to wear masks, improved hygiene, and voluntary participation in social distancing, along with other formal policy interventions not included in our study.

COVID-19 models identified the presence of a statistically significant positive association between changes in residents’ consumer mobility patterns and county-level confirmed COVID-19 case growth rates. In a similar manner to prior studies, we created a feature to represent changes in residents’ consumer mobility patterns, specifically the percent change in aggregated daily POS transaction counts compared to pre-pandemic baseline days. Using data from the resident counties of our two samples (i.e., Suffolk County, MA and Washtenaw County, MI) we uncovered a statistically significant positive association between 12-day lagged changes in consumer mobility and daily confirmed COVID-19 case growth rates. Specifically, a 10% percent decrease in the 12-day lagged change in consumer mobility was associated with a 3.2% decrease in the daily confirmed COVID-19 case growth rate. This aligns with the findings of previous studies that lagged changes in human mobility patterns were associated with COVID-19 transmission, most notably at a lag of 9–12 days [25,26,27,29,30,31]. However, our measure of mobility, daily POS transaction counts, displayed a slightly weaker association with COVID-19 spread in comparison to the mobility metrics used in other studies.

The COVID-19 modeling portion of our study also possesses a few limitations. First, daily POS credit card transaction counts are a limited human mobility measure, as it is impossible to determine the amount of human contact associated with each transaction. Second, there exist other potential COVID-19 mitigation factors beyond mobility reduction (e.g., mask wearing, hand washing, and case tracing) that our analysis does not consider due to a lack of data. Third, the COVID-19 case data used in our analysis may contain errors due to reporting issues and limited testing capacity common among many medical facilities during the initial disease outbreak. Fourth, available data only allowed us to control for the number of tests performed at the state, rather than county, level.

## 7. Conclusions

US state governments relied on a complex combination of NPIs to control the initial outbreak of COVID-19. NPIs led to varied patterns of human movement and behavioral changes throughout the country. Simultaneously, the progression and intensity of the COVID-19 pandemic fluctuated considerably by location. In conjunction, these two factors make measuring the impact of NPIs on consumer mobility patterns and the spread of COVID-19 a non-trivial task, which explains the popularity of studies aimed at achieving this end. The results of our study support prior research indicating that NPIs, specifically containment measures such as workplace closures, stay-at-home requirements, and gathering restrictions, effectively reduced human mobility during the initial COVID-19 outbreak. Additionally, our results reinforce previous findings that the effect of subsequent changes in mobility patterns on disease transmission were not perceptible for 9–12 days. However, unlike previous studies which often uncovered stronger relationships between changes in human mobility patterns and the spread of COVID-19, our results indicate that daily POS transactions are a limited mobility metric. Ultimately, our findings are tied to a study of large random samples of individuals residing in two cities (i.e., Boston, MA and Ann Arbor, MI) during roughly the first three months following the declaration of COVID-19 as a national emergency. Future studies should seek to analyze consumers’ mobility patterns across various states over a longer duration.

## Figures and Tables

**Figure 1 ijerph-21-00067-f001:**
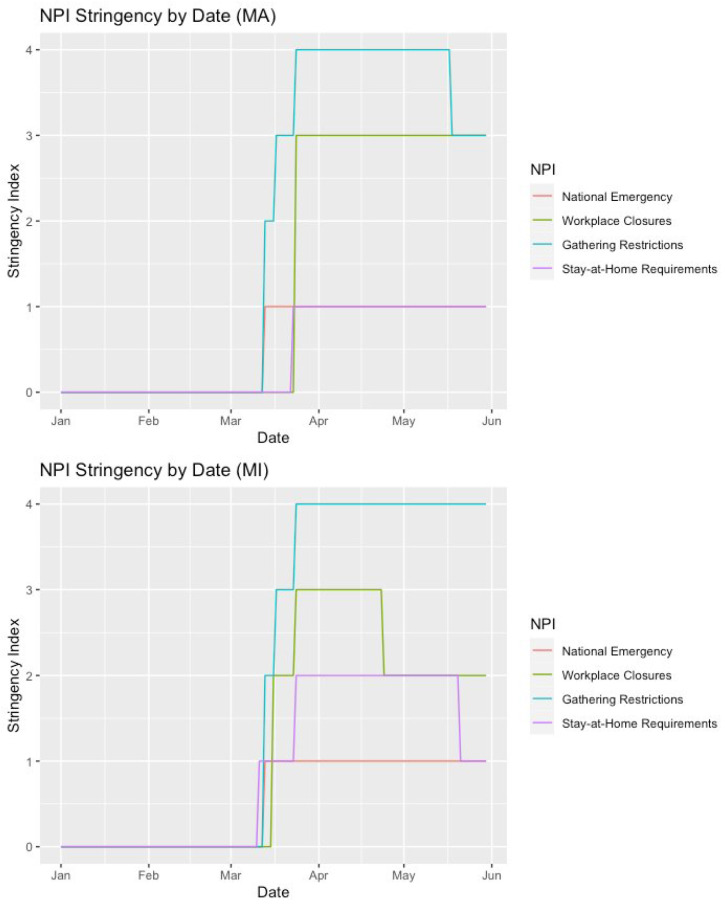
Non-pharmaceutical intervention stringency by date (MA and MI).

**Figure 2 ijerph-21-00067-f002:**
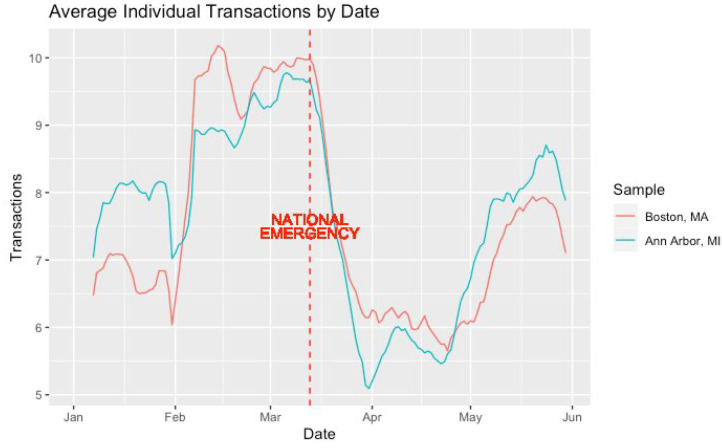
Average individual POS transaction count by date.

**Figure 3 ijerph-21-00067-f003:**
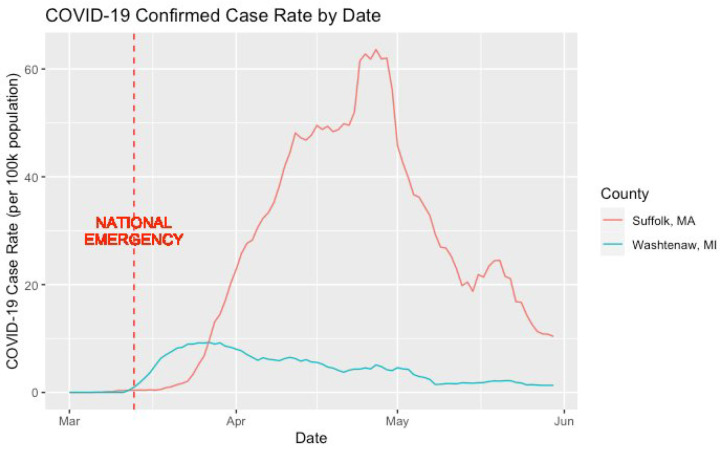
Confirmed COVID-19 case rate (per 100k population) by date.

**Figure 4 ijerph-21-00067-f004:**
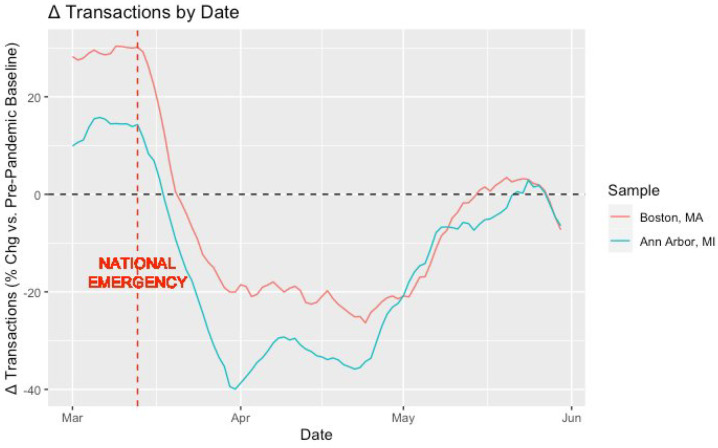
Δ Transactions (percentage change vs. pre−pandemic baseline) by date.

**Table 1 ijerph-21-00067-t001:** US state-level policy data.

Policy	Description	Stringency
National Emergency	On 13 March, President Trump officially declared COVID-19 a US national emergency.	0—not effective
		1—effective
Workplace Closures	A record of work closings. Workplace closures began in MA on 19 March and MI on 16 March.	0—no measurement
		1—recommended
		2—required (some)
		3—required (all)
Stay-At-Home Requirements	A record of orders requiring people to “shelter-in-place” and otherwise confine to home. Stay-at-home requirements for non-essential workers began in MA on 23 March and MI on 11 March.	0—no measurement
		1—recommended
		2—required (some exceptions)
		3—required (minimal exceptions)
Gathering Restrictions	A record of the cut-off size for bans on gatherings. Gathering restrictions began in both MA and MI on 13 March.	0—no measurement
		1—restrict >1000 people
		2—restrict 101–1000 people
		3—restrict 11–100 people
		4—restrict ≤10 people

**Table 2 ijerph-21-00067-t002:** Proportion of days by NPI stringency 1 January–31 May (MA and MI).

NPI	Stringency	Days (MA)	Days (MI)
Workplace Closure	None	83 (55.0%)	75 (49.7%)
	Recommended	0 (0.0%)	0 (0.0%)
	Required (some)	0 (0.0%)	45 (29.8%)
	Required (all)	68 (45.0%)	31 (20.5%)
Stay-at-Home Reqs.	None	82 (54.3%)	70 (46.4%)
	Recommended	69 (45.7%)	23 (15.2%)
	Required (some ex.)	0 (0.0%)	58 (38.4%)
	Required (min ex.)	0 (0.0%)	0 (0.0%)
Gathering Restrictions	None	72 (47.7%)	72 (47.7%)
	Restrict >1000	0 (0.0%)	0 (0.0%)
	Restrict 101–1000	4 (2.7%)	4 (2.7%)
	Restrict 11–100	20 (13.2%)	7 (4.6%)
	Restrict <10	55 (36.4%)	68 (45.0%)

**Table 3 ijerph-21-00067-t003:** Model (1) results (MA and MI residents).

NPI (MA)	Fixed Effects	$e^{\beta }$	95% CI	*p*-Value	NPI (MI)	Fixed Effects	$e^{\beta }$	95% CI	*p*-Value
National Emergency	Intercept	2.28	(2.18,2.39)	0.00 ***	National Emergency	Intercept	3.18	(3.07,3.30)	0.00 ***
	Effective	0.90	(0.88,0.93)	0.00 ***		Effective	0.76	(0.75,0.78)	0.00 ***
	Weekend	0.81	(0.80,0.82)	0.00 ***		Weekend	0.77	(0.77,0.78)	0.00 ***
Workplace Closure	Intercept	2.35	(2.24,2.45)	0.00 ***	Workplace Closure	Intercept	3.24	(3.12,3.36)	0.00 ***
	Recommended	NA	NA	NA		Recommended	NA	NA	NA
	Required (some)	NA	NA	NA		Required (some)	0.86	(0.84,0.88)	0.00 ***
	Required (all)	0.83	(0.80,0.86)	0.00 ***		Required (all)	0.58	(0.57,0.60)	0.00 ***
	Weekend	0.81	(0.80,0.82)	0.00 ***		Weekend	0.77	(0.76,0.78)	0.00 ***
Stay-at-Home Reqs.	Intercept	2.33	(2.23,2.44)	0.00 ***	Stay-at-Home Reqs.	Intercept	3.22	(3.10,3.34)	0.00 ***
	Recommended	0.85	(0.83,0.98)	0.00 ***		Recommended	0.93	(0.90,0.96)	0.00 ***
	Required (some ex.)	NA	NA	NA		Required (some ex.)	0.70	(0.68,0.71)	0.00 ***
	Required (min ex.)	NA	NA	NA		Required (min ex.)	NA	NA	NA
	Weekend	0.80	(0.80,0.81)	0.00 ***		Weekend	0.77	(0.76,0.78)	0.00 ***
Gathering Restrictions	Intercept	2.31	(2.21,2.42)	0.00 ***	Gathering Restrictions	Intercept	3.24	(3.12,3.37)	0.00 ***
	Restrict >1000	NA	NA	NA		Restrict >1000	NA	NA	NA
	Restrict 101–1000	1.05	(1.00,1.11)	0.04 *		Restrict 101–1000	1.14	(1.08,1.19)	0.00 ***
	Restrict 11–100	0.91	(0.87,0.94)	0.00 ***		Restrict 11–100	0.92	(0.88,0.96)	0.00 ***
	Restrict <10	0.85	(0.83,0.88)	0.00 ***		Restrict <10	0.70	(0.69,0.72)	0.00 ***
	Weekend	0.81	(0.80,0.82)	0.00 ***		Weekend	0.77	(0.76,0.78)	0.00 ***

Notes: * p<0.1, *** p<0.01; We were unable to acquire estimates for a few stringency levels of the various NPIs due to a lack of data.

**Table 4 ijerph-21-00067-t004:** Model (Equation 2) results (MA and MI).

Lag Period	Fixed Effects	β	95% CI	*p*-Value
08 Days	Intercept	20.40	(6.48,34.33)	0.00 ***
	Δ Transactions	0.18	(−0.09,0.44)	0.19
	Test Rate	−0.09	(−0.16,−0.01)	0.02 **
	Weekend	−28.38	(−41.73,−15.03)	0.00 ***
10 Days	Intercept	20.03	(6.60,33.46)	0.00 ***
	Δ Transactions	0.18	(−0.08,0.45)	0.18
	Test Rate	−0.08	(−0.15,0.00)	0.04 **
	Weekend	−30.27	(−43.73,−16.82)	0.00 ***
12 Days	Intercept	18.04	(6.85,29.23)	0.00 ***
	Δ Transactions	0.32	(0.08,0.57)	0.01 **
	Test Rate	−0.05	(−0.13,0.02)	0.14
	Weekend	−26.14	(−39.23,−13.04)	0.00 ***
14 Days	Intercept	20.40	(6.48,34.33)	0.00 ***
	Δ Transactions	0.18	(−0.09,0.44)	0.19
	Test Rate	−0.09	(−0.16,−0.01)	0.02 **
	Weekend	−28.38	(−41.73,−15.03)	0.00 ***

Notes: ** p<0.05, *** p<0.01

## Data Availability

All computer code used to carry out our statistical analysis is publicly available via GitHub: https://github.com/JosephZuccarelli/Publications (accessed on 15 June 2023). Restrictions apply to the availability of the full dataset under analysis. Data were obtained from BDEX and are available from the authors with the permission of BDEX.

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
