# Peer review of "Assessing the Impact of Non-Pharmaceutical Interventions on Consumer Mobility Patterns and COVID-19 Transmission in the US"

_ijerph, 2024, doi:10.3390/ijerph21010067_

Round 1

Reviewer 1 Report

Comments and Suggestions for Authors

1. log(transactionsij) = β0 + βNPIi + βweekendi + γ0j . (1)

The subscripts used on the right and the left sides do not match. If j is reserved for individuals then t could be used for denoting time on both sides, and i could be reserved for other purposes. If you use ijt to capture different dimensions, it will solve your problem. Your model is alright, but the notations indicating the dimensions are not quite clear to us. 

2. Literature is OK. The research gap is fine. The idea is nice.

3. Using Poisson mixed-effects modeling is also fine.

4. From your study it is not quite clear why you have chosen Boston and Ann Arbor as a comparison. Please clarify this point for us.

5. This is a good work. I wish you good luck.

Reviewer 2 Report

Comments and Suggestions for Authors

Line 36-37: You mentioned "...the number of COVID-19 cases worldwide had grown above 9800, including over 200 deaths". I suggest you specify 'the number of confirmed/diagnosed COVID-19 cases' and '200 confirmed deaths'. The actual number of cases was obviously much higher.

Line 51-6: please cite. "Adherence to social distancing recommendations is pertinent to reducing the spread of COVID-19, as COVID-19 is not only highly contagious, but also infects many individuals without showing any common symptoms of the virus. Unfortunately, however, voluntary social distancing guidelines are not sufficient to end the pandemic, as many people may choose not to adhere to these recommendations. Therefore, it is crucial that governments take concrete action to limit close-contact between many people"

Under the 'related work' section, I suggest you give examples from Saudi Arabia. Saudi Arabia implemented one of the most successful NPI measures to control the pandemic.

Your introduction and literature review section is too long. Please revise to reduce the length.

line 65-66: reference needed. ". Governors across all fifty states chose to implement these different forms of NPIs at various points over time, while learning from each other in the process"

Please highlight the unique contributions of your study focusing on policy implications.

In conclusion, you mentioned 'Future studies should seek to analyze more complete measures of consumer mobility patterns across various states over a longer duration.' What did you mean by 'more complete measures'. Please explain in your article.

Reviewer 3 Report

Comments and Suggestions for Authors

It is an interesting topic which is rarely addressed.

However, there are many aspects that need to be improved:

Lines 27-67: Your text provides a comprehensive overview of the early stages of the COVID-19 pandemic, including the initial identification of the virus, its spread, and the response measures taken by governments, especially in the United States. The information seems accurate and aligns with the timeline of events as of my last knowledge update in January 2022. However, it's important to note that developments in the understanding of COVID-19 and the response to the pandemic may have occurred since then. If there have been significant changes or updates, you might want to verify and incorporate the latest information. Additionally, consider breaking down longer paragraphs into smaller ones for better readability. This can make the information more digestible for readers. Overall, your text effectively communicates the key events and responses related to the early stages of the COVID-19 pandemic.

Lines 68-80 Suggestions:

•Consider providing a brief explanation of why POS transactions are considered moments of high virus exposure, as not all readers may be familiar with this concept.

•You may want to break down the last sentence into two for better readability and emphasis. For example, you could have a sentence dedicated to summarizing the analysis approach and another for highlighting the practical implications of the results.

•If relevant, include information on the timeframe of the study. Readers might be interested in understanding the period covered by the data.

•Depending on your audience, you might consider adding a sentence about the significance or broader context of the study. Why is understanding the impact of these specific NPIs on consumer mobility important? What implications could it have for public health policies?

Lines 82-104 Suggestions:

•Consider breaking down the last sentence into two or restructuring it for improved flow. The transition from discussing the OxCGRT project to specific containment and closure policies could be smoother;

•Maintain consistency in your terminology. In the beginning, you mention "23 indicators of government response," but later, you specify four types of policies. It might be helpful to clarify that these four types are derived from the 23 indicators.

Lines 105-134 Suggestions:

•Be consistent in how you refer to the authors in your citations. For instance, you could consistently use either full names or initials throughout the passage;

•Consider breaking down the last sentence into two for improved clarity and flow. This separation can help the reader transition more smoothly to the next part of your discussion;

•Why are these particular studies relevant to your investigation, and how do they contribute to the understanding of the impact of NPIs?

Lines 135-157: Here are a few suggestions:

•  Maintain consistency in how you refer to the authors in your citations. For example, you could consistently use either full names or initials throughout the passage;

•  Consider using transition phrases to guide the reader through the various studies and their findings. This can enhance the overall flow of the paragraph;

•  When presenting quantitative results, be clear about what the percentages represent. For instance, when mentioning a "9-10% increase in the rate at which state residents remained in their homes full time," clarify whether this is a percentage increase or a relative change.

Lines 158-188 Suggestions:

•When presenting percentages, consider specifying whether they represent relative changes or percentage points. For instance, when mentioning a "7-45% reduction in mobility after twenty days," clarify whether this is a relative change from baseline or a percentage point decrease;

•Consider condensing some sentences for brevity while maintaining clarity. For example, the sentence "In this study, we expand upon previous research through the analysis of a novel, individual-level consumer mobility dataset that documents the daily POS credit card transaction counts of two large random samples of urban residents between January 1st - May 31st, 2020" could be streamlined for conciseness.

Lines 191-198 Suggestions:

•Consider rephrasing the first sentence to enhance clarity. For example, you might say, "The OxCGRT project systematically collected information on policy measures implemented by US government leaders to curb the spread of COVID-19."

 •While you mention the period from March 1st through May 31st, 2020, it could be helpful to reiterate this information or mention it explicitly in the first sentence. For example, you could say, "For each state, OxCGRT used publicly available sources to record data on indicators of government response to the outbreak of COVID-19 dating back to the beginning of 2020, with a specific focus on the period from March 1st through May 31st [9]."

•Table 1: Consider briefly summarizing or referring to Table 1 in the text to provide a preview of the NPIs you're focusing on. This can help readers understand what specific measures are being discussed.

Lines 201-212 Suggestions:

•Ensure consistency in date formats. For example, you use "January 1st - February 29th, 2020" but later mention "May 31st, 2020." It's good to keep the format consistent throughout.

Lines 214-224 Suggestions:

•Mention the specific time period for which the COVID-19 case data was collected. This can help readers understand the temporal scope of your analysis.

•Briefly explain the rationale for selecting Suffolk County, Massachusetts, and Washtenaw County, Michigan, for your study. This can provide context for why these specific counties were chosen.

•Provide a bit more detail or context about why you chose to calculate the daily exponential COVID-19 case growth rate and how this measure aligns with the objectives of your study.

Lines 226-233 Suggestions:

•Mention any specific challenges or limitations related to the availability of testing data at the county level during the early phases of the pandemic. This can help contextualize your decision to use state-level testing data.

•Provide a concise explanation of why county-level population totals were integrated into the statistical analysis. How does this weighting account for demographic differences, and why is it important for your study?

Lines 235-249 Suggestions:

•Elaborate on what is meant by "different stringency levels" concerning the implementation of NPIs. This could help readers better understand the variations in the impact of different interventions.

•Include a brief rationale for choosing linear and generalized linear mixed-effects models with autoregressive covariance structures. Why were these specific modeling approaches chosen, and how do they suit the characteristics of your dataset?

•Clarify what is meant by "Consumer Mobility Modeling subsection" and "COVID-19 Modeling subsection." A brief explanation or preview of the content within these subsections could be beneficial.

Lines 251-267 Suggestions:

•While you briefly mention one-hot encoding, you could provide a bit more clarity on how this technique is applied and what it means in the context of your study. This can enhance understanding, especially for readers less familiar with statistical modeling.

•Consider providing a brief explanation of terms like β0, βNPIi, βweekendi, and γ0j. While you've defined the response variable, transactions ij, offering a brief description or interpretation of other terms could be beneficial.

Lines 269-290 Suggestions:

•While you briefly mention the explanatory variable of interest in your COVID-19 models, it might be helpful to explicitly state what this variable represents. For example, you could say, "The explanatory variable of interest in our COVID-19 models was the percentage change in consumer mobility patterns."

•Provide a clear definition or explanation of what is considered a "pre-pandemic baseline day." You mention that it's the median value of the sample’s daily POS transaction count, but specifying this in a bit more detail could be beneficial.

•Did you perform sensitivity analyses or consider different lag times?

Lines 300-310 It appears that this is a mixed-effects regression model designed to analyze the relationship between changes in consumer mobility patterns and COVID-19 case growth rates. 

In equation (2), consider providing a brief explanation of each variable and coefficient. For example, you could include a small legend or annotations to clarify the meaning of terms like 1β1, 2β2, 3β3, and the response variable case growth rate ij. This can enhance understanding for readers not familiar with the specific notation.

Results

Lines 313-325 Suggestions:

•In Figure 1, consider adding a brief legend or labels to specify the stringency levels on the y-axis. This can help readers interpret the stringency of NPIs over time more easily.

•For Table 2, provide a key or brief explanation of what each stringency level represents. This will help readers understand the significance of the proportions presented in the table.

Lines 327-345 Suggestions:

•In the last paragraph, you mention the exponentiation of the NPI coefficient terms. You could provide a bit more explanation or context on what these coefficients represent. For example, discuss how a positive or negative coefficient might indicate an increase or decrease in daily POS transactions associated with the implementation of a specific NPI.

•Consider including a visual representation of the patterns described in Figure 2, perhaps a line graph showing the average individual POS transaction count over time. This can enhance the reader's understanding and engagement with the data.

Lines 346-369 Suggestions:

•Ensure consistency in the terminology used throughout. For instance, you mentioned "recommended stay-at-home orders" and "required stay-at-home orders (with some exceptions)." Clarify the distinction between these terms for better understanding.

Modeling

Lines 371-397 Suggestions:

•Consider explicitly defining the variables in your model equation for readers who might not be familiar with the notation. For example, provide a brief explanation of what each variable represents in the context of your study.

•Maintain consistency in the terminology you use. For example, you mentioned "COVID-19 case rate" in the first sentence, and later referred to "daily COVID-19 case growth rate." Ensure that you use consistent terms to describe the outcome variable.

Discussion

Lines 399-428 Suggestions:

•Consider providing a brief introduction to the BDEX dataset, explaining its uniqueness and why it was chosen for your analysis. This can help readers understand the significance of the dataset in the context of your study.

•Structure your explanation in a way that clearly delineates different aspects of your study, such as dataset characteristics, consumer mobility models, and findings related to NPIs.

•Explicitly state key findings in a concise manner. This can include highlighting the most significant associations, such as the strong negative relationship between workplace closures and POS transactions.

Lines 429-475: You acknowledge the limitations, including the nature of daily POS transaction counts as a limited measure of human mobility and the potential influence of other COVID-19 mitigation factors that were not considered due to data constraints.This transparency about the strengths and limitations of your study enhances the credibility and reliability of your findings.

Your study reinforces the understanding that changes in mobility patterns may take around 9-12 days to manifest in disease transmission.The acknowledgment of the limitations of daily POS transactions as a mobility metric adds nuance to your results, emphasizing the need for a more comprehensive understanding of consumer mobility patterns.

Limitations of the study are also mentioned in lines 455-463.

I think that in lines 464-475 an attempt was made to form some conclusions, but these should be mentioned more clearly through a separate paragraph.

My comments are only intended to make the paper better. Good luck!
